# Cefotaxime-, Ciprofloxacin-, and Extensively Drug-Resistant *Escherichia coli* O157:H7 and O55:H7 in Camel Meat

**DOI:** 10.3390/foods12071443

**Published:** 2023-03-29

**Authors:** Khalid Ibrahim Sallam, Yasmine Abd-Elrazik, Mona Talaat Raslan, Kálmán Imre, Adriana Morar, Viorel Herman, Hanan Ahmed Zaher

**Affiliations:** 1Department of Food Hygiene and Control, Faculty of Veterinary Medicine, Mansoura University, Mansoura 35516, Egypt; jasmineabd556@gmail.com (Y.A.-E.);; 2Food Hygiene Department, Animal Health Research Institute (AHRI), Dokki, Giza, Cairo 12618, Egypt; monaraslan66@gmail.com; 3Department of Animal Production and Veterinary Public Health, Faculty of Veterinary Medicine, University of Life Sciences “King Mihai I” from Timișoara, 300645 Timișoara, Romania; adrianamo2001@yahoo.com; 4Department of Infectious Diseases and Preventive Medicine, Faculty of Veterinary Medicine, University of Life Sciences “King Mihai I” from Timişoara, 300645 Timișoara, Romania

**Keywords:** camel meat, Shiga toxins, virulence genes, antimicrobial resistance, PCR

## Abstract

The present study aimed to explore for the first time the occurrence and the antimicrobial resistance profiles of *E. coli* O157:H7 and O55:H7 isolates in camel meat in Egypt. Among the 110 camel meat samples examined using standardized microbiological techniques, 10 (9.1%) and 32 (29.1%) were positive for *E. coli* O157:H7 and *E. coli* O55:H7, respectively. In total, 24 isolates were verified as *E. coli* O157:H7, while 102 isolates were confirmed serologically as *E. coli* O55:H7. Multiplex PCR revealed the existence of *eaeA*, *stx1*, *stx2*, and EHEC-*hlyA* among *E. coli* O157:H7 and O55:H7 isolates (*n* = 126) at various percentages. According to their resistance against 14 antibiotics, 16.7% and 83.3% of O157:H7 isolates and 8.6% and 76.5% of O55:H7 isolates were classified into extensively drug-resistant and multi-drug-resistant, respectively, whereas 29.4% and 22.2% of *E. coli* isolates were resistant to cefotaxime and ciprofloxacin, respectively. The study results emphasize that camel meat may be a vehicle for multi- and extensively drug-resistant *E. coli* O157:H7 and O55:H7 strains, indicating a potential threat to public health. Further studies based on the molecular evidence of the antimicrobial resistance genes and enrolling a larger number of samples are recommended for a better understanding of the antimicrobial resistance phenomenon of camel-meat-originating pathogenic *E. coli* strains.

## 1. Introduction

Enterohemorrhagic *Escherichia coli* (EHEC) O157 serotype is a well-documented human foodborne pathogen able to produce diarrhea, hemorrhagic colitis (HC), and hemolytic uremic syndrome (HUS) in consumers in both developing and industrialized countries [1]. The main elements associated with the EHEC O157 pathogenicity include Shiga toxins 1 and 2 (encoded by *stx1* and *stx2* genes), intimin (encoded by *eae* gene), and EHEC hemolysin (encoded by Ehly gene). Most of the Shiga-toxin-producing *Escherichia coli* (STEC) O157 isolates have *eae* and *stx2* genes [2].

Although the EHEC O157:H7 have been incriminated in most outbreaks of HC and HUS, several outbreaks caused by STEC serotype O55:H7 have been recorded in many countries. Outbreaks of HUS caused by STEC serotype O55:H7 occurred in England between 2014 and 2018 [3]. Sporadic outbreaks with *E. coli* O55:H7 have been reported in the Czech Republic, Italy, and Germany [4]. In July 2014, in Dorset County, England, 31 cases were linked to the O55:H7 outbreak: 13 (42%) of those patients suffered from HUS, with 8 (66%) of them having neurologic complications and 11 (90%) necessitating long-term treatment for kidney transplantation [5]. Again, in England, in 2017, in the county of Surrey, O55:H7 was found as the implicated organism in human infection in two people and suspected in a further five [6]; additionally, in 2018, in Leicestershire, O55:H7 was reported in two children who developed HUS, terminating in death [7].

*E. coli* O55:H7 strains have a chromosome backbone very similar to that of EHEC O157:H7 [8,9]. It has been presumed that the usual STEC O157:H7 clone developed from the enteropathogenic *E. coli* serotype O55:H7. Various studies have suggested the evolutionary models predicting the gradual acquirement of a Shiga toxin (Stx)-encoding bacteriophage of the enteropathogenic *E. coli* O55:H7 ancestor strain with subsequent replacement of the *rfb* locus encoding the antigen of the somatic O55, with that encoding the O157 antigen [8,9,10].

Food of animal origin is considered one of the most important sources of STEC infection in humans. Among the domesticated ruminants, cattle are recognized as major natural reservoirs for the STEC and play important roles in the dissemination of such pathogen to humans through contaminated animal-origin foodstuffs [11,12].

Egypt’s camel meat production in 2019 was 8664 metric tons, which puts the country at number 12 in the rankings among camel-meat-producing countries in the world. Egypt is overtaken by Niger, which was ranked number 11 with 10,725 metric tons. Saudi Arabia ranked the highest with 108,679 metric tons, while Kenya, Somalia, and the United Arab Emirates ranked at numbers 2, 3, and 4, respectively.

Camel meat has an exceptional feature, as it contains low fat, good-quality protein, and cholesterol contents with moderate amounts of polyunsaturated fatty acids. Camel meat can transmit diseases and infections through consumption even though the foodborne illness-outbreak rate linked with the ingestion of camel meat is somewhat low. The consumption of unsafe raw or undercooked meat products such as ground beef and beef-burger hamburger has been frequently associated with several human infections [11]. Contamination of ruminant carcasses, including camels, with STEC occurs throughout slaughtering procedures during the hide removal or the evisceration process as well as during butchering, which enables the organisms to be entirely mixed into the meat, especially when minced into beef burgers. Skinning, evisceration, and carcass splitting, if occurring on the abattoir floor, lead to high opportunities for meat contamination from the skin, intestines, and polluted ground. Several studies published from many countries including Egypt have demonstrated the contamination of ruminant meat and derived products with *E. coli* O157:H7 and O55:H7 [13,14,15,16].

Antimicrobial resistance has become a significant threat to human and animal health worldwide and hinders the usage of conventional antibiotics for the treatment of specific infections. Food-producing animals generally require antimicrobials as a treatment or prophylactic course against various bacterial infections. The over- and misuse of antimicrobials at a worldwide level induced the emergence of multidrug-resistant bacterial strains, which disseminate in human food of animal origin and constitute a great threat to human health. Nowadays, the increased speed of the emergence of multidrug-resistant bacterial pathogens constitute one of the greatest challenges in the management of human as well as animal infections. Moreover, the frequently occurring treatment failures reflect the necessity of new, potent, and safe antimicrobial agents.

Until recently, almost no studies have been available about the prevalence of *E. coli* O157:H7 and O55:H7 in camel meat in Egypt. Therefore, the present study aimed to investigate the prevalence of *E. coli* O157:H7 and *E. coli* O55:H7 strains in the marketed camel meat distributed in camel-butcher shops in Beheira Province, Egypt, and to explore their antimicrobial resistance profile against 14 antimicrobials from different categories. Additionally, the presence of certain virulence markers in the isolated strains including *eae, stx2, stx1*, and *hlyA* genes were evaluated.

## 2. Materials and Methods

### 2.1. Collection of Samples

A total of 110 camel meat samples taken from 110 different camel carcasses were collected throughout November 2021 to March 2022 from different camel-butcher shops distributed in Beheira Governorate, Egypt. The samples (500 g each) were taken from the thigh region and were individually and aseptically packaged into a plastic bag, labeled with a number, and transferred in an icebox to the Laboratory of Food Hygiene and Control Department, Faculty of Veterinary Medicine, Mansoura University, where the microbiological examinations were performed.

### 2.2. Isolation and Identification of E. coli Strains

A total of 25 g from each harvested sample was aseptically homogenized with 225 mL of modified tryptone soya broth (CM0989; Oxoid-Thermo Fisher Scientific Inc., St. Leon Roth, Germany) containing vancomycin at a final concentration of 40 mg/mL. The resultant homogenate was subsequently incubated at 37 °C for 18 h. Next, the enriched culture was streaked onto sorbitol Mac Conkey agar (CM0813; Oxoid-Thermo Fisher Scientific Inc.) previously supplemented with cefixime and potassium tellurite (SR0172; Oxoid-Thermo Fisher Scientific Inc.). The plates were then incubated for 24 h at 37 °C and the grown presumptive *E. coli* O157:H7 and *E. coli* O55:H7 colorless colonies (sorbitol nonfermenting) were selected. Five to ten colonies were taken and sub-cultured onto nutrient agar (CM0003; Oxoid-Thermo Fisher Scientific Inc.) slopes for further characterization. A total of 280 colonies (colorless) were exposed to biochemical tests designed for *E. coli* identification, including indole, methyl red, Voges–Proskauer, and citrate tests. Typical *E. coli* O157:H7 and *E. coli* O55:H7 isolates exhibited positive reactions for indole production and methyl red tests as well as negative reactions for each of the Voges–Proskauer and citrate utilization tests. The enterohemolytic phenotype of the strains isolated was determined through the streaking of *E. coli* colonies on blood agar plates that contained 5% washed sheep erythrocytes, followed by incubation at 37 °C. The occurrence of a narrow hemolysis zone within 18–24 h clarifies a positive result [17].

### 2.3. Serological Identification of E. coli Isolates

*E. coli* O157:H7 and *E. coli* O55:H7 isolates were serologically characterized by using a rapid diagnostic *E. coli* antisera kit (Denka Seiken Co., Ltd., Tokyo, Japan) designed for diagnosis of the enteropathogenic serotypes, according to the manufacturer instructions. The serological characterization was performed at the Food Analysis Center, Faculty of Veterinary Medicine, Benha University, Egypt.

### 2.4. Molecular Characterization of E. coli Isolates

#### 2.4.1. Isolation of Genomic DNA

The genomic DNA of *E. coli* O157:H7 and *E. coli* O55:H7 isolates as well as from the positive (*E. coli* O157:H7 reference strains) and negative (*E. coli* K12 DH5α) controls was isolated using the commercially available DNA extraction kits (Roche Applied Science, Basel, Switzerland), according to the manufacturer’s recommendations.

#### 2.4.2. PCR for Detection of rfbO157 Gene Specific for *E. coli* O157

The biochemically confirmed *E. coli* strains were further tested by PCR for identification of the *rfb_O157_* marker gene of *E. coli* O157 using the specific primer set: F: 5’-GCGGAACAAAACCATGTGCA-3’ and R: 5’-ACTGGCCTTGTTTCGATGAG-3’, which was previously constructed by Sallam et al. [13] to yield an 800 bp DNA size specific for the *rfb*_O157_ gene. A twenty-five-microliter reaction mixture comprising 12.5 μL of DreamTaq Green Master Mix (Thermo Scientific, St. Leon Roth, Germany), 1 μL from each of the forward and reverse primer (10 pmol each), 1 μL of the extracted genomic DNA as a template, and 9.5 μL RNase free water added to the 25 μL final volume. The PCR amplification condition included an initial denaturation step at 96 °C for 4 min, followed by 30 cycles of 96 °C for 30 s, 60 °C for 45 s, and 72 °C for 90 s. The final cycle was followed by an extension step of 72 °C for 5 min. The SimpliAmp Thermal Cycler (Thermo Fisher Scientific Inc.) was used for PCR reactions.

#### 2.4.3. Detection of *stx1*, *stx2*, *eaeA*, and *hylA* Virulence Genes with PCR

*E. coli* O157:H7 and *E. coli* O55:H7 strains (*n* = 126) isolated from camel meat were examined for the evidence of some virulence genes specific for pathotyping of the diarrheagenic *E. coli* as *eaeA*, *stx1*, *stx2*, and EHEC-*hlyA*. The primer sequences constructed for PCR reactions of the *stx1* (614 bp) and *stx2* (779 bp) genes, according to Dhanashree and Mallya [18], were 5’-acactggatgatctcagtgg-3’ for *stx1* forward, 5’-ctgaatccccctccattatg-3’ for *stx1* reverse, 5’-ccatgacaacggacagcagtt-3’ for *stx2* forward, and 5’-cctgtcaactgagcagcactttg-’3 for *stx2* reverse. Moreover, the constructed sequences of the primer set used for PCR amplification of each gene were as follows: 5’-gtggcgaatactggcgagact-3’ for sense and 5’-ccccattctttttcaccgtcg-3’for antisense to yield an 890-bp DNA fragments for *eaeA* [19], whereas the primer set sequences for PCR amplification of *hylA* gene were 5’-acgatgtggtttattctgga-’3 for sense and 5’-cttcacgtgaccatacatat-’3 for antisense to yield a DNA size of 165 bp [20].

Multiplex PCR of 50 μL volume containing 25 μL of DreamTaq Green Master Mix (Thermo Scientific, St. Leon Roth, Germany); 1 μL from each the sense and antisense primer (10 pmol each) specific for each of the *stx1*, *stx2*, *eaeA*, and *hylA* virulent genes (8 μL in total); 2.5 μL of *E. coli* genomic DNA as a template; and 14.5 μL RNase-free water added to a final volume of 50 μL. Multiplex PCR for *stx1*, *stx2*, *eaeA*, and *hylA* genes consisted of an initial denaturation step for 3 min at 95 °C, followed by 35 cycles of denaturation at 95 °C for 20 s, annealing at 58 °C for 30 s, and polymerization at 72 °C for 90 s. The final cycle was followed by 72 °C for 5 min as an extension step.

An *E. coli* O157:H7 Sakai reference strain was employed as a positive control (positive for *stx1*, *stx2*, *eaeA*, and *hylA*), while *E. coli* K12 DH5α was used as a negative control non-pathogenic strain not harboring any virulence gene. The resulted amplicons (6 μL aliquots from each reaction mixture) were separated within the agarose gel (1.5%) electrophoresis at 100 V for 50 min and stained with ethidium bromide solution for 25 min. The agarose gel was then subjected to an ultra-violet (UV) transilluminator for visualization and photographing the separated PCR products.

### 2.5. Antimicrobial Resistance Profile and Multiple Antibiotic Resistance (MAR) index of E. coli O157:H7 and E. coli O55:H7 Isolated Strains

The antimicrobial susceptibility testing for the isolated *E. coli* O157:H7 and *E. coli* O55:H7 strains was performed with the disk diffusion susceptibility technique applied according to the guidelines specified by the Clinical and Laboratory Standards Institute [21]. Fourteen antibiotics susceptibility discs (Oxoid Limited, Basingstoke, Hampshire, UK) from nine antimicrobial classes were used with the following variable concentrations: aminoglycosides: amikacin (30 μg), gentamicin (10 μg); carbapenemes: imipenem (10 μg); cephalosporins: cefotaxime (30 μg), ceftriaxone (30 μg), cephalothin (30 μg); fluoroquinolones: levofloxacin (5 μg), ciprofloxacin (5 μg); Macrolides: erythromycin (15 μg); lincomycins: clindamycin (10 μg); penicillins: ampicillin (10 μg), penicillin (10 IU); sulfonamides: sulfamethoxazole (25 μg); and tetracyclines: tetracycline (30 μg). The strains tested were categorized into susceptible, intermediate, and resistant. The MAR index for each isolated strain was calculated according to the formula set by Singh et al. [22] as follows:$$\mathrm{MAR}\;\mathrm{index}=\frac{\mathrm{The}\;\mathrm{number}\;\mathrm{of}\;\mathrm{resistant}\;\mathrm{isolates}}{\mathrm{Total}\;\mathrm{number}\;\mathrm{of}\;\mathrm{tested}\;\mathrm{antibiotics}.}$$

## 3. Results and Discussion

### 3.1. Prevalence of the Isolated E. coli Strains in Camel Meat Samples

Among the 110 camel meat samples tested, 10 (9.1%) samples were positive for *E. coli* O157:H7, which was present either alone (*n* = 4) or coexisting with *E. coli* O55:H7 (*n* = 6), while 32 (29.1%) camel meat samples were found positive for *E. coli* O55:H7, which existed either alone (*n* = 26) or as a mixed contamination (*n* = 6) with *E. coli* O157:H7 strains (Figure 1). Data about the prevalence of *E. coli* O157:H7 and/or *E. coli* O55:H7 in camel meat are very limited. A study conducted on 50 camel meat samples in Iran [23] indicated that only one (2.0%) sample was contaminated with *E. coli* O157. Likewise, Hassan et al. [24] could not detect the enteropathogenic *E. coli* in any of the 15 camel meat samples examined in Qaliubiya Governorate, Egypt, although they could detect it in 13.3% (2/15), 13.3% (2/15), and 33.3% (5/15) of camel spleen, liver, and kidney samples, respectively. In the fecal sample taken from healthy camels, however, *E. coli* O157:H7 was isolated from 19 (19%) of 100 camels’ fecal specimens collected from 100 apparently healthy camels in the middle of Iraq [25], while it was present in 3.3% from the fecal samples taken from 140 healthy camels at a slaughterhouse in the United Arab Emirates [26].

The relatively higher incidence of *E. coli* O157:H7 (9.1%) and *E. coli* O55:H7 (28.6%) among the tested camel meat samples in this study may be attributed to the cross-contamination of such organisms that normally inhabitant the gut of food animals including camels and are shed out with the animal feces, and therefore, it is very likely that *E. coli* can pollute the meat during the slaughtering process, especially if a faulty evisceration takes place, and also from the contaminated hide. In this context, Elder et al. [27] revealed a significant association between fecal and hide prevalence of EHEC O157 and cattle carcass contamination by such organisms.

### 3.2. Molecular Characterization and Virulence Genes Distribution among E. coli O157:H7 and E. coli O55:H7 Isolated from Camel Meat

Serological identification of *E. coli* isolates indicated that 102 isolates were serotyped as O55:H7, while only 24 isolates were serotyped as O157:H7. The twenty-four *E. coli* O157:H7 isolates were further confirmed by PCR as positive for the existence of the *rfb_O157_*-specific gene, which was detected at 800 bp size (Figure 2a). Both *E. coli* O157:H7 and *E. coli* O55:H7 isolates (*n* = 126) were examined by multiplex PCR for the existence of *eaeA*, *stx1*, *stx2,* and EHEC-*hlyA*, which were identified at the expected molecular sizes of 890 bp, 614 bp, 779 bp, and 165 bp, respectively (Figure 2b).

The association of the four virulent genes determined in the 126 *E. coli* isolates obtained from camel meat in the present investigation varied among the isolates.

Based on the presence of these four representative virulence genes, *E. coli* O157:H7 (*n* = 24) were categorized into four groups (A–D), while *E. coli* O55:H7 (*n* = 102) were categorized into seven groups (I–VII) (Table 1). Interestingly, groups A (*n* = 13), B (*n* = 6), C (*n* = 3), and D (*n* = 2) of O157:H7 isolates exhibited the same existence pattern of the virulence gene in comparison to groups V (*n* = 5), II (*n* = 21), I (*n* = 46), and III (*n* = 12) of O55:H7 isolates, respectively. Precisely, groups A and V harbored *stx1*, *stx2*, *eaeA*, and *hlyA*; groups B and II harbored *stx2*, *eaeA*, and *hlyA*; group C and I harbored *stx2* and *eaeA*, while group D and III harbored *stx1*, *stx2*, and *eaeA* genes (Table 1). The comparable pattern of the virulence genes distribution in both *E. coli* O157:H7 and *E. coli* O55:H7 isolates is not surprising since *E. coli* O55:H7 and *E. coli* O157:H7 strains have a very similar chromosome backbone [7,8], and many studies have indicated that the STEC O157:H7 clone developed from *E. coli* O55:H7.

#### 3.2.1. Shiga Toxin Genes (*stx1* and *stx2*)

Shiga toxin 2 (*stx2*) was the most prevalent virulence gene detected in STEC O157:H7 and O55:H7 isolated from camel meat in the current investigation, where it was identified in 97.6% (123/126) of the isolates. On the other hand, only 27.8% (35/126) of the isolates harbored *stx1* (Table 1). Among the 123 *stx2*-positive *E. coli* O157:H7 and O55:H7 strains tested in the present study, 32 (25.4%) strains harbored both *stx1* and *stx2* genes, while 91 (72.2%) strains harbored *stx2* while lacking *stx1* (Table 1). It was previously pointed out that the strains harboring the *stx2* gene are possibly more severe than those harboring the *stx1* gene or even the strains containing both *stx1* and *stx2* [28,29]. Studies have revealed that *stx2* is the most significant virulence element linked with severe illnesses and that *stx2*-carrying strains are more commonly correlated with HUS than the strains carrying *stx1* [30,31]. Actually, *stx2* is described to be 1000 times more cytotoxic compared to *stx1* towards the micro-vascular endothelial cells of human kidneys [32].

The existence rate along with the association of *stx1* and *stx2* genes in STEC strains from food of animal origin varied from one study to another. Lee et al. [33], in Korea, found that 64% of the STEC strains obtained from fresh beef carried *stx2*, whereas 14% carried both *stx1* and *stx2*. Beutin et al. [34], in Germany, identified *stx2* in 81% and *stx1* in 40% of the STEC strains recovered from fresh meat samples. Mora et al. [35], in Spain, showed that 28 (29%) of the 96 STEC isolates recovered from minced beef possessed *stx1* genes, 49 (51%) carried *stx2* genes, and 19 (20%) carried both *stx1* and *stx2*. Slanec et al. [36], in another study in Germany, reported that 70.7%, 9.3%, and 20.0% of the 75 STEC strains isolated from animal origin foodstuffs carried *stx2*, *stx1*, and both *stx1* and *stx2*, respectively. Additionally, Eid et al. [37] found that all of the three (100%) instances of *E. coli* O55:H7 obtained from broiler chicken in Egypt were positive for *stx1*, while two (66.7%) of them were positive for the *stx2* gene. Contrary to our results, Rahimi et al. [23] revealed that the sole *E. coli* O157 isolate recovered from camel meat in Iran was lacking *stx1* and *stx2* virulent genes. Even if the striking findings of the virulence pattern of the isolated *E. coli* strains in the present study can be considered suggestive, strengthening the occurrence of positive results recorded in other surveys with different detection rates, further investigations that focus on the evidence of other virulence genes/factors (e.g., *elt*, *est*, *daa*D, *inv*E, *Eagg*, or *astA* genes) and enrolling a larger number of isolates are still necessary for a comprehensive understanding of the pathogenicity of the camel meat origin *E. coli* strains.

#### 3.2.2. Intimin (*eaeA*) and Enterohemolysin (*hlyA*) Genes

The *eaeA* is the gene that encodes the intimin adherence factor, which is considered an outer membrane protein necessary for the intimate attachment of *E. coli* to the intestinal mucosa of the host. The human pathogenic *E. coli* isolates are mostly positive for the *eae* gene. In the present study, the intimin (*eaeA*) gene was the second most prevalent virulence gene after *stx2*, where it was detected in 85.7% (108/126) of the isolates. On the other hand, 43.7% (55/126) of the isolates harbored *hlyA* genes (Table 1). The *eaeA* gene existed in all of the 24 (100%) *E. coli* O157:H7 and in 84 (82.4%) of the 102 *E. coli* O55:H7 isolates isolated from the camel meat investigated in this study. Likewise, Sallam et al. [13], in Egypt, indicated that 93.3% (14/15) of the *E. coli* O157 strains recovered from beef products were positive for the *eaeA* gene. Similarly, Cagney et al. [38] identified the *eae* gene in 95.3% (41/43) of *E. coli* O157:H7 isolates obtained from meat products in Ireland. Moreover, Chapman et al. [39] detected the *eae* gene in all (100%) of the 72 *E. coli* O157:H7 strains recovered from raw beef and lamb products in the UK. Conversely, a low existence rate of 26% (25/96) was determined for the *eae* gene in STEC strains from minced beef in Spain [35], while a much lower prevalence rate of 5% was found among the isolated STEC strains recovered from samples of fresh meat harboring *eaeA* genes [34]. Nonetheless, Eid et al. [37] found that one (33.3%) of the three *E. coli* O55:H7 strains recovered from broiler chicken in Egypt was positive for the *eaeA* gene, while Barlow et al. [40] failed to detect the *eaeA* gene in the any of the meat-products-originating STEC strains (*n* = 184) obtained in Australia. A strong association has been clarified by several publications between the presence of the *eae* gene and the competency of STEC isolates to induce severe infections in humans, especially HUS and bloody diarrhea [28,41,42]. In the present study, the association of *eaeA* and *stx2* genes was determined in all (100%) of the 24 *E. coli* O157:H7 and in 84 (82.4%) of the 102 *E. coli* O55:H7, while the association of the *eaeA* gene and *stx1* was determined in 15 (62.5%) of the 24 *E. coli* O157:H7 and in 17 (16.7%) of the 102 *E. coli* O55:H7. Lee et al. [33] found that 7% and 14% of the STEC strains recovered from beef harbored both *stx1* and *eae* and both *stx2* and *eae*, respectively.

The EHEC-*hlyA* is another principal agent implicated in the virulence of *E. coli*. In the present survey, 19 (79.2%) of the 24 *E. coli* O157:H7 and 36 (35.3%) of the 102 *E. coli* O55:H7 isolates were positive for EHEC-*hlyA* gene, with a general prevalence of 43.7% (55/126) among the 126 STEC isolates tested (Table 1). EHEC-hlyA gene was detected at a higher prevalence rate of 93.3% (14/15) among the isolated *E. coli* O157 strains from beef products in Egypt [13]. A study conducted on beef and mutton meat in Hamedan, Iran, indicated that the single *E. coli* O157:H7 isolate as well as the nine (9/57; 15.8%) non-O157:H7 isolated strains were positive for the *hlyA* gene [43]. A lower detection rate of 25% (2/8) was also detected for the *hlyA* gene among the STEC strains obtained from cattle meat in Iran [16]. Similarly, a lower positive rate of 18.2% (2/11) for *hlyA* was determined in *E. coli* O55:H7 isolated from different meat products in Saudi Arabia [14]. Contrary to our findings, Rahimi et al. [23] revealed that the sole *E. coli* O157 isolate from camel meat in Iran was deficient in *eaeA* and *ehlyA* virulent genes. In the same country, the markedly genetic resistance of the human-originating *E. coli* strains was recently demonstrated in a study conducted by Jomehzadeh et al. [44].

The most frequent profile of the virulence gene detected in *E. coli* isolates from camel meat was *stx2* and *eaeA*. Therefore, our isolated strains should be considered pathogenic for humans since the capability of STEC to induce severe illness and their potential to cause outbreaks of infection are associated with *stx2*-positive strains.

### 3.3. Antimicrobial Resistance Profile and MAR Index of E. coli O157:H7 and E. coli O55:H7 Strains

Antimicrobial-resistant bacteria are considered a serious problem that has received significant attention worldwide due to their ability to obstruct the treatment of severe infections in human patients [45]. In the present study, all (100%) of the obtained *E. coli* O157:H7 and *E. coli* O55:H7 isolates displayed absolute resistance to at least three antibiotics from different classes, namely clindamycin (lincomycins class), penicillin (penicillins class), and erythromycin (macrolydes class) (Table 2), which are commonly used in the veterinary medicine.

Furthermore, very high resistance rates of 92.9%, 71.4%, and 48.4% was observed regarding tetracycline, ampicillin, and sulfamethoxazole-trimethoprim, respectively (Table 2). Tetracyclines and beta-lactam antibiotics such as ampicillin are still frequently used, especially in livestock treatment in Egypt [46]. The recorded high resistance towards tetracycline, ampicillin, and sulfamethoxazole-trimethoprim in this survey was consistent with previous studies on *E. coli* strains isolated from beef in Egypt [47,48]. Furthermore, the aminoglycosides, including amikacin and gentamicin, are extensively used in animal husbandry as broad-spectrum antibiotics and as growth promoters. The recovered *E. coli* O157:H7 and *E. coli* O55:H7 isolates displayed a resistance rate of 39.7% and 16.7% towards amikacin and gentamicin, respectively (Table 2). In a previous study in Ethiopia, 100% of *E. coli* O157:H7 isolated from raw cattle meat were susceptible to amikacin and gentamicin [49].

Third-generation cephalosporin antibiotics such as cefotaxime and ceftriaxone have been widely used in the management of infections caused by Gram-negative bacteria [50]. In the present study, 84.9%, 29.4%, and 3.97% of the identified *E. coli* O157:H7 and *E. coli* O55:H7 isolates were resistant to cephalothin (first-generation cephalosporin), cefotaxime, and ceftriaxone (Table 2), respectively. On the contrary, all *E. coli* O157:H7 isolates from cattle in Tunisia were sensitive to cefotaxime [51]. Our results are in line with those reported in a previous study conducted in Italy by Grispoldi et al. [52], who found that 89.7% of *E. coli* from bovine lymph nodes were resistant to cephalothin, which is recognized as an effective antimicrobial against Gram-negative as well as Gram-positive bacteria. Furthermore, a relatively lower resistance rate of 57% was observed in *E. coli* serovars isolated from bovine origin meat samples (four cows and three buffaloes) in Egypt against cephalothin [53]. Interestingly, despite the two third-generation cephalosporins, namely ceftriaxone and cefotaxime, being similar in their antibacterial spectrum, indications, and route of administration, the isolates examined showed a resistance rate of 29.4% against cefotaxime and a resistance rate of only 3.97% against ceftriaxone (Table 2), which may be attributed to the difference in the administration dose and elimination half-life between the two antibiotics (ceftriaxone: 2 g once a day with half-life elimination of 8.8 h versus cefotaxime: 2 g every 4 h, 1.2 h) [54,55]. Likewise, Gums et al. [56] revealed that the nonmeningeal *Streptococcus pneumoniae* isolates were more susceptible to ceftriaxone than to cefotaxime. The higher resistance rate of the identified *E. coli* O157:H7 and *E. coli* O55:H7 isolates in the present study against cefotaxime constitutes a potential threat for public health and indicates the crucial requirement for the cautious administration of antibiotics in the veterinary medicine.

Fluoroquinolones are extensively used against a wide variety of Gram-positive and Gram-negative bacteria. In this study, the identified *E. coli* O157:H7 and *E. coli* O55:H7 isolates (*n* = 126) revealed high resistance rates of 22.2% and 8.7% toward ciprofloxacin (second-generation quinolone) and levofloxacin (third-generation quinolone), respectively (Table 2), which is in agreement with the resistance rates found in a previous study conducted in Saudi Arabia by El-Ghareeb et al. [57], who found 17.6% of *E. coli* isolates obtained from minced camel meat were resistant to ciprofloxacin. In contrast, Sabala et al. [46], in Egypt, found that all of the *E. coli* isolates (*n* = 150) from raw beef samples were susceptible to ciprofloxacin. The resistance rates of the identified *E. coli* O157:H7 and *E. coli* O55:H7 isolates (*n* = 126) against imipenem, which is recognized as a first-line drug in the treatment of human infections, were 5.56%. Conversely, none of the examined *E. coli* isolates (*n* = 150) from raw beef samples were resistant to carbapenem antimicrobials (imipenem and meropenem) [46].

Interestingly, 16.7% (4/24) and 83.3% (20/24) of the identified *E. coli* O157 isolates in the present study were categorized, based on their resistance profiles against the 14 different antimicrobials tested, into extensively drug-resistant (XDR) and multi-drug resistant (MDR), respectively, with a mean MAR index of 0.497 (Table 3), while 4.9%, 18.6%, and 76.5% of the identified *E. coli* O55:H7 isolates (*n* = 102) were categorized, according to their resistance profiles toward the 14 different antibiotics examined, into pan-drug resistant (PDR), extensively drug-resistant (XDR), and multi-drug resistant (MDR), respectively, with a mean MAR index of 0.527 (Table 3). Surprisingly, 3.97% (5/126) of *E. coli* isolates (all belong to *E. coli* O55:H7 serovars) exhibited resistance towards all of the 14 tested antibiotics, with a MAR index equal to 1.0 (Table 3). This finding could constitute a potential threat to public health. The average MAR index for *E. coli* O157:H7 and O55:H7 isolates in the present study was 0.514 (Table 3). *E. coli* isolates with a MAR index of >0.2 indicate a misuse and overuse of antibiotics, while *E. coli* isolates with a MAR index of 0.4 and above indicate human fecal contamination [58]. Therefore, it is crucial to establish monitoring systems that impose a rational usage of antibiotics in veterinary medicine for protecting public health from the spreading of multidrug-resistant bacteria to humans via animal-origin foodstuffs.

Altogether, the recorded overall antimicrobial resistance pattern of the isolated camel-meat-originating pathogenic *E. coli* strains can be considered worrisome in the present study, which largely agrees with results of other studies carried out in other regions of the world. These results strengthen the fact that the antimicrobial resistance phenomenon has become a significant threat to human and veterinary medicine. However, the recorded differences in terms of susceptibility towards the tested antimicrobials can be markedly influenced by the number of enrolled samples, study design, the used testing methodologies, and guidelines in interpreting the results, etc. Furthermore, supplementary studies focusing on continuous monitoring for the presence of pathogenic *E. coli* strains in camel meat combined with the testing of their resistance towards a wider variety of antimicrobials are still recommended.

## 4. Conclusions

The results of the present survey highlighted that the camel meat marketed in Beheira Governorate, Egypt, is contaminated with multi- and extensively drug-resistant Shiga toxigenic *Escherichia coli* (STEC) O157:H7 and O55:H7 strains, with an average MAR index for the isolates of 0.497 and 0.527, respectively, which emphasizes the necessity to optimize the use of the antibiotics in human and veterinary medicine. However, for a better understanding of the antimicrobial resistance phenomenon, in the case of the camel-meat-originating *E. coli* pathogenic strains, further investigations based on the screening of the antimicrobial resistance genes in a larger number of samples are still required. The study also elucidated that the most frequent profile of the virulence gene detected in *E. coli* O157:H7 and O55:H7 isolates from camel meat included *stx2* and *eaeA*, indicating that the isolated strains have the potential to cause severe infection in humans.

## Figures and Tables

**Figure 1 foods-12-01443-f001:**
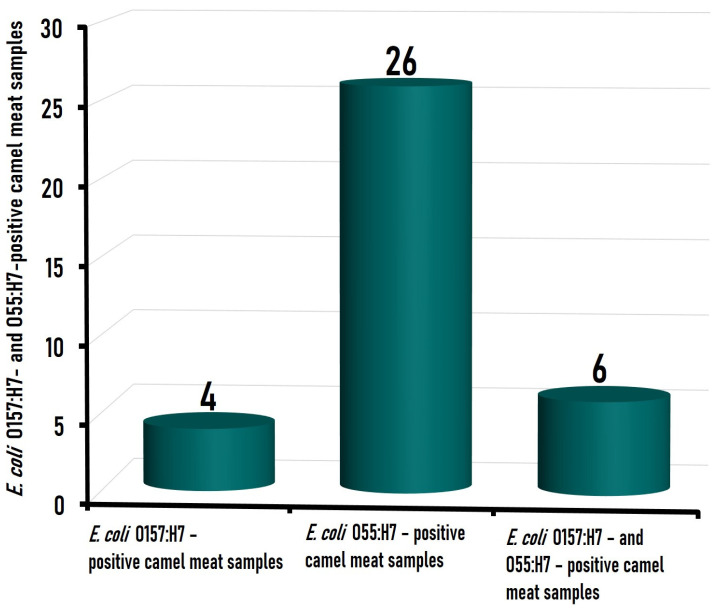
Number of camel meat samples contaminated with *Escherichia coli* O157:H7 and *Escherichia coli* O55:H7, from the total of 110 investigated.

**Figure 2 foods-12-01443-f002:**
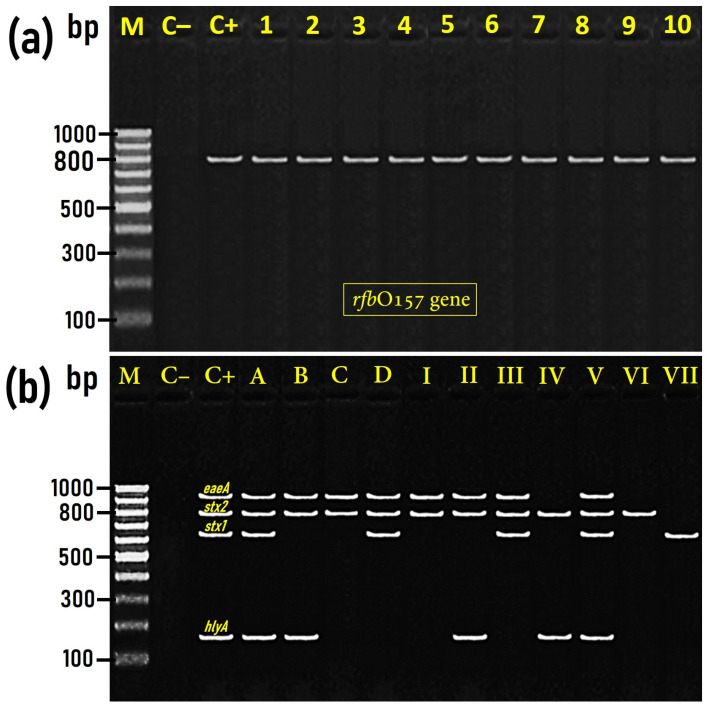
Representative agarose gel electrophoresis image for PCR amplicon of the *rfb*O157 gene (800 bp) characteristic for *Escherichia coli* O157:H7 (**Panel** (**a**)) and for the multiplex PCR products of *stx1* (614 bp), *stx2* (779 bp), *eaeA* (890 bp), and *hlyA* (165 bp) virulence genetic markers in both *E. coli* O157:H7 and *E. coli* O55:H7 (**Panel** (**b**)). M, 100 bp ladder as a molecular DNA marker; Lane C–, PCR runs resulted from using a template of *E. coli* K12 DH5α genome as a negative control strain (does not harbor any virulence gene); Lane C+, control positive *E. coli* O157:H7 Sakai as positive reference strain for *rfb*O157 gene (**a**) and *stx1*, *stx2*, *eaeA*, and *hlyA* genes (**b**). Lanes 1–10 representative isolates for the 10 O157:H7-positive camel meat samples. Lanes A (O157:H7, group A; 13 isolates positive for *stx1*, *stx2*, *eaeA*, and *hlyA* genes); Lane B (O157:H7, group B; 6 isolates positive for *stx2*, *eaeA*, and *hlyA* genes); Lane C (O157:H7, group C; 3 isolates positive for *stx2* and *eaeA* genes); Lanes D (O157:H7, group D; 2 isolates positive for *stx1*, *stx2*, and *eaeA* genes). Lane I (O55:H7, group I; 46 isolates positive for *stx2* and *eaeA* genes); Lane II (O55:H7, group II; 21 isolates positive for *stx2*, *eaeA*, and *hlyA* genes); Lane III (O55:H7, group III; 12 isolates positive for *stx1, stx2*, and *eaeA* genes); Lane IV (O55:H7, group IV; 10 isolates positive for *stx2* and *hlyA* genes); Lane V (O55:H7, group V; 5 isolates positive for *stx1*, *stx2*, *eaeA*, and *hlyA* genes); Lane VI (O55:H7, group VI; 3 isolates positive only for *stx2* gene); Lane VII (O55:H7, group VII; 3 isolates positive only for *stx1* gene).

**Table 1 foods-12-01443-t001:** Distribution of virulence genes among the obtained *E. coli* O157:H7 and O55:H7 isolates (*n* = 126).

Serotype	Group	Number of *E. coli* Isolates	Virulence Genes			
			*stx1*	*stx2*	*eaeA*	*hlyA*
O157:H7	A	13	+	+	+	+
	B	6	−	+	+	+
	C	3	−	+	+	−
	D	2	+	+	+	−
O55:H7	I	46	−	+	+	−
	II	21	−	+	+	+
	III	12	+	+	+	−
	IV	10	−	+	−	+
	V	5	+	+	+	+
	VI	5	−	+	−	−
	VII	3	+	−	−	−
Total		126	35	123	108	55

Note: *stx1*, Shiga-toxin 1 gene; *stx2*, Shiga-toxin 2 gene; *eaeA*, intimin gene; *hylA*, hemolysin gene.

**Table 2 foods-12-01443-t002:** Antimicrobial susceptibility of *E. coli* O157:H7 and O55:H7 strains (*n* = 126).

No	Antimicrobial Agent (Acronym)	Sensitive		Intermediate		Resistant	
		NO	%	NO	%	NO	%
1	Clindamycin (CL)	–	–	–	–	126	100
2	Penicillin (P)	–	–	–	–	126	100
3	Erythromycin (E)	–	–	–	–	126	100
4	Tetracycline (T)	7	5.56	2	1.59	117	92.9
5	Cephalothin (CN)	15	11.9	4	3.17	107	84.9
6	Ampicillin (AM)	29	23.0	7	5.56	90	71.4
7	Sulfamethoxazole (SXT)	60	47.6	5	3.97	61	48.4
8	Amikacin (AK)	68	53.97	8	6.35	50	39.7
9	Cefotaxime (CF)	79	62.7	10	7.94	37	29.4
10	Ciprofloxacin (CP)	98	77.8	–	–	28	22.2
11	Gentamicin (G)	100	79.4	5	3.97	21	16.7
12	Levofloxacin (L)	109	86.5	6	4.76	11	8.7
13	Imipenem (IPM)	115	91.3	4	3.17	7	5.56
14	Ceftriaxone (C)	121	96.03	–	–	5	3.97

Note: CL, clindamycin; P, penicillin; E, erythromycin; T, tetracycline; CN, cephalothin; AM, ampicillin; SXT, sulfamethoxazole; AK, amikacin; CF, cefotaxime; CP, ciprofloxacin; G, gentamicin, L, levofloxacin; IPM, imipenem; C, ceftriaxone.

**Table 3 foods-12-01443-t003:** Classification of *E. coli* O157:H7 and O55:H7 (*n* = 126) based on their antimicrobial resistance degree against 14 antimicrobials tested.

*E. coli* Strains	Number of Isolates	Antimicrobial Resistance Profile	Classes with Resistance	Mar Index	Classification of Strains	
						No. and (%)
O157:H7 (*n* = 24)	2	CL, P, E, T, CN, AM, SXT, AK, CF, CP, G, L, IPM	Macrolides, lincomycins, penicillins, tetracyclines, cephalosporins, aminoglycosides, sulfonamides, fluoroquinolones, carbapenemes	0.929	Extensively drug-resistant	4 (16.7%)
	2	CL, P, E, T, CN, AM, SXT, AK, CF, CP, G	Macrolides, lincomycins, penicillins, tetracyclines, cephalosporins, aminoglycosides, sulfonamides, fluoroquinolones	0.786		
	2	CL, P, E, T, CN, AM, SXT, AK, CF	Macrolides, lincomycins, penicillins, tetracyclines, cephalosporins, aminoglycosides, sulfonamides	0.643	Multidrug-resistant	20 (83.3.0%)
	4	CL, P, E, T, CN, AM, SXT, AK	Macrolides, lincomycins, penicillins, tetracyclines, cephalosporins, aminoglycosides, sulfonamides	0.571		
	5	CL, P, E, T, CN, AM	Macrolides, lincomycins, penicillins, tetracyclines, cephalosporins, aminoglycosides	0.429		
	6	CL, P, E, T	Macrolides, lincomycins, penicillins, tetracyclines	0.357		
	3	CL, P, E	Macrolides, lincomycins, penicillins	0.214		
	Sum 24	Average MAR Index for O157:H7		0.497		
O55:H7 (*n* = 102)	5	CL, P, E, T, CN, AM, SXT, AK, CF, CP, G, L, IPM, C	Aminoglycosides, carbapenemes, cephalosporins, fluoroquinolones, macrolides, lincomycins, penicillins, sulfonamides, tetracyclines	1.000	Pan-drug resistant	5 (4.9%)
	4	CL, P, E, T, CN, AM, SXT, AK, CF, CP, G, L	Macrolides, lincomycins, penicillins, tetracyclines, cephalosporins, aminoglycosides, sulfonamides, fluoroquinolones	0.857	Extensively drug-resistant	19 (18.6%)
	8	CL, P, E, T, CN, AM, SXT, AK, CF, CP, G	Macrolides, lincomycins, penicillins, tetracyclines, cephalosporins, aminoglycosides, sulfonamides, fluoroquinolones	0.786		
	7	CL, P, E, T, CN, AM, SXT, AK, CF, CP	Macrolides, lincomycins, penicillins, tetracyclines, cephalosporins, aminoglycosides, sulfonamides, fluoroquinolones	0.714		
	7	CL, P, E, T, CN, AM, SXT, AK, CF	Macrolides, lincomycins, penicillins, tetracyclines, cephalosporins, aminoglycosides, sulfonamides, fluoroquinolones	0.643	Multidrug-resistant	78 (76.5%)
	9	CL, P, E, T, CN, AM, SXT, AK	Macrolides, lincomycins, penicillins, tetracyclines, cephalosporins, aminoglycosides, sulfonamides	0.571		
	11	CL, P, E, T, CN, AM, SXT	Macrolides, lincomycins, penicillins, tetracyclines, cephalosporins, aminoglycosides, sulfonamides	0.500		
	24	CL, P, E, T, CN, AM	Macrolides, lincomycins, penicillins, tetracyclines, cephalosporins, aminoglycosides	0.429		
	17	CL, P, E, T, CN	Macrolides, lincomycins, penicillins, tetracyclines, cephalosporins	0.357		
	4	CL, P, E, T	Macrolides, lincomycins, penicillins, tetracyclines	0.286		
	6	CL, P, E	Macrolides, lincomycins, penicillins,	0.214		
	Sum 102	Average MAR Index for O55:H7 isolates		0.527		
**Total**	126	Average MAR index for O157:H7 and O55:H7 isolates	Type of resistance	0.514		

Note: CL, clindamycin; P, penicillin; E, erythromycin; T, tetracycline; CN, cephalothin; AM, ampicillin; SXT, sulfamethoxazole; AK, amikacin; CF, cefotaxime; CP, ciprofloxacin; G, gentamicin, L, levofloxacin; IPM, imipenem; C, ceftriaxone.

## Data Availability

The data presented in this study are available on request from the corresponding author.

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
