# Peer review of "Cefotaxime-, Ciprofloxacin-, and Extensively Drug-Resistant Escherichia coli O157:H7 and O55:H7 in Camel Meat"

_foods, 2023, doi:10.3390/foods12071443_

Round 1

Reviewer 1 Report

Dear authors. Congratulations for your research. It is well proposed, methods are well described and results are concisely expressed and supported by your findings. Recent studies carried out in my research group (and others) have revealed meat as a strong vehicle for antibacterial resistances spread. This is a worrisome concern due to the extremely serious implications of bacterial resistances development over global health, due to the impact over some infectious diseases treatment. Nevertheless, although there are some data available, specific studies about less common meat producing animal such as camel are not so often, even less including molecular biology techniques. Nevertheless, here you have some tips to even improve it.

Author Response

Reviewer #1:

Dear authors. Congratulations for your research. It is well proposed, methods are well described and results are concisely expressed and supported by your findings. Recent studies carried out in my research group (and others) have revealed meat as a strong vehicle for antibacterial resistances spread. This is a worrisome concern due to the extremely serious implications of bacterial resistances development over global health, due to the impact over some infectious diseases treatment. Nevertheless, although there are some data available, specific studies about less common meat producing animal such as camel are not so often, even less including molecular biology techniques. Nevertheless, here you have some tips to even improve it.

Dear reviewer, our sincere thanks for taking the time to review ours manuscript, and your close attention to detail. We highly appreciate your overall positive feed-back regarding the quality of the submission! Please see below for our responses to your comments:

Line 19, 22, 102, 175, 240, 358, 384, etc.: Unify expressing numbers with symbols or letters.

Done according to the reviewer suggestion, except the lines indicating numbers up to ten, where their writhing in letters were kept in the original form „e.g. L 240 or L358 from the submitted version”

Line 38: Have been recorded

Corrected according to the reviewer suggestion.

Line 61: Metric Tones should go with capital letters?

Corrected in „metric tones”. We apologize for this beginner mistake!

Line 82. Not only in developing countries, it is a worldwide problem. It might be even more worrisome there, but in case you specify that, you should provide specific data or references.

Corrected according to the reviewer suggestion!

Line 94: Here you mention 110 samples whilst you talk about 112 in the abstract. Also in results section.

The correct number of the total processed samples is 110. It was corrected. We apologize for this.

Line 109: Non-fermenting. Double space.

The correction was done.

Lines 160-161: you refer to seconds as s and sec. It is just “s”. Figure 1. I propose changing the graph and, instead including absolute values, including percentages.

The correction was done. Concerning the Fig. 1 and with respect to the reviewer requirement the authors want to keep the submitted version of the Figure 1, but at the figure caption, to be more informative, they added „...from the total of 110 investigated” to be more informativ.

Line 280. You make an extremely good discussion, but it might be good including a final conclusion of it summarizing the results of other authors and yours, and although you include a joint summary of sections 3.2.1 and 3.2.2 at the end of section 3.2.2, I miss the same in both genes included in section 3.2.2. The same for the AMR sections.

This reviewer request was approached within a combined action. Thus, according to the reviewer requirement, and also to complete the study limitations requested by another reviewer, at the end of the mentioned subheading chapters, two solid paragraphs were inserted in the revised version of the manuscript as follows: „Even if the yielded striking findings of the virulence pattern of the isolated E. coli strains in the present study can be considered suggestive, strengthening the occurrence of positive results recorded in other surveys with different detection rates, further investigations, focusing on the evidence of other virulence genes/factors (e.g. elt, est, daaD, invE, Eagg, or astA genes) enrolling a larger number of isolates are still necessary, for a comprehensive understanding of the pathogenicity of camel meat origin E. coli strains.”

and

„Altogether, the recorded overall antimicrobial resistance pattern of the isolated camel meat origin pathogenic E. coli strains can be considered worrisome in the present study, which largely agree with results of other studies carried out in other regions of the world. These results strengthen the fact that the antimicrobial resistance phenomenon became a significant threat to human and veterinary medicine. However, the recorded differences in terms of susceptibility towards the tested antimicrobials can be markedly influenced by the number of enrolled samples, study design, the used testing methodologies, and guidelines in interpreting the results, etc. Furthermore, supplementary studies, focusing on the continuous monitoring fo the presence of pathogenic E. coli strains in camel meat combined with the testing of their resistance towards a wider variety of antimicrobials are still recommended.”

Line 326. Considers

The correction was done!

Line 333. Tetracycine should start with capital letters to match the rest of antibiotics.

The correction was done!

Lines 348-355: Please, unify the way you write Gram, with or without capital letters.

The correction was done throughout the manuscript with Capital letter!

Line 366. Cefotaxime. There is a comma where it should go a point.

The correction was done!

Line 386: MAR is previously described.

„(multiple antibiotic resistance)” – after it first appearance in the text was deleted and corrected throughout the manuscript!

Line 411: Comma after respectively. References: Complete reference 7 Unify DOIs (some of them are underlined, some others not, some others half-half, etc.)

The comma was added. For the reference no. 7 the accession data was added. The DOIs were unified throughout the reference list.

Line 532. Some spaces lack in the name of the journal. Revise the format of the whole list of references.

The authors tried to do their best to standardize and revise the format of the reference list in agreement with the journal's requirements! However, the authors hope that the possible remained non-conformities will be solved the professional MDPI team during the manuscript proof correction!

THANK YOU AGAIN FOR YOUR REVIEW!

Reviewer 2 Report

- The present study has a significant influence, but it needs a major revision.

- Please write the scientific names of bacterial pathogens and genes in the correct form all over the manuscript and in the References section (should be italic).

-The English of the manuscript should be reviewed and syntax and errors should be corrected before publication.

Abstract:
- The abstract must illustrate the used methods and the most prevalent results (give more hints about methods and results). Besides, rephrase the aim of the work and the main conclusion of your findings.

Introduction: (it needs to be more informative):

- The authors should illustrate the public health importance concerning the emergence of multidrug-resistant (MDR) bacterial pathogens that reflect the necessity of new potent and safe antimicrobial agents.

-Rephrase the aim of the work to be clear and better sound.

Material and methods:

- Isolation and identification of E. coli strains:
• Discuss the methods of isolation and identification of E. coli. Besides, specific references should be added.
- Antimicrobial susceptibility testing:
• Add the names of the antimicrobial classes of the tested antibiotics.

- The correlation between phenotypic and genotypic multidrug resistance should be performed.
-Specific references should be added to all the used methods and techniques.

-Results:

-Add this subtitle: Phenotypic characteristics of the recovered isolates
• Illustrate the phenotypic characteristics of the recovered E. coli isolates.
-Discuss more on significant results of this study.
-Antimicrobial susceptibility testing:
• Illustrate in a new table the occurrence of MDR (Multidrug resistance) among the recovered isolates as the following (illustrate the names of the antimicrobial classes and different antibiotics):
No. of strains/Type of resistance
(Antimicrobial classes and different antibiotics). The antibiotic -resistance genes
-The correlation (Correlation coefficient) between phenotypic and genotypic multidrug resistance should be performed.

-You must support your results with illustrating figures.

-Discussion:

- Discuss more on the alarming results of this research. 
- The authors are advised to illustrate the real impact of their findings.
-Illustrate the different mechanisms of antimicrobial resistance in E. coli.
Use the following valuable studies performed on pathogen isolates in the discussion section and add related references, including:

https://doi.org/10.1007/s11033-022-07215-5

-Conclusion
- Should be rephrased to be sounded. A real conclusion should focus on the question or claim you articulated in your study. Which resolution has been the main aim of your paper?

Author Response

Reviewer #2:

- The present study has a significant influence, but it needs a major revision.

Dear reviewer, our sincere thanks for taking the time to review our manuscript, and your close attention to details. We highly appreciate your overall positive feed-back regarding the quality of our submission! Please see below our responses to your comments:

- Please write the scientific names of bacterial pathogens and genes in the correct form all over the manuscript and in the References section (should be italic).

During the revision process, the authors tried to do their best to fulfill this requirement! In this regard, the authors hope that the possible remained non-conformities will be solved with the help of the professional MDPI team during the manuscript proofreading!

-The English of the manuscript should be reviewed and syntax and errors should be corrected before publication.

The English content of the manuscript has been carefully revised with the help of a native English speaker, resulting in a significant improvement.

Abstract:

- The abstract must illustrate the used methods and the most prevalent results (give more hints about methods and results). Besides, rephrase the aim of the work and the main conclusion of your findings.

In agreement with the reviewer requirement, we filled the missing and requested data from the abstract section.

Introduction: (it needs to be more informative):

- The authors should illustrate the public health importance concerning the emergence of multidrug-resistant (MDR) bacterial pathogens that reflect the necessity of new potent and safe antimicrobial agents.

According to the reviewer requirement, the introduction section was completed with the following sentences “Nowadays, the increased speed of the emergence of multidrug-resistant bacterial pathogens constitute one of the greatest chalenge in the management of human, as well as animal infections. Moreover, the frequently occurred treatment failures reflect the necessity of the new potent and safe antimicrobial agents.”

-Rephrase the aim of the work to be clear and better sound.

Rephrased as requested!

Material and methods:

-Isolation and identification of E. coli strains:

  • Discuss the methods of isolation and identification of E. coli. Besides, specific references should be added.

The isolation and identification of the E. coli strains was described in the original submitted version of the manuscript. According to the reviewer requirement at the end of the paragraph the requested reference was added (reference no. 17).

- Antimicrobial susceptibility testing:

  • Add the names of the antimicrobial classes of the tested antibiotics.

As the reviewer requested, the tested antimicrobials were presented according to the antimicrobial classes to which they belongs.

- The correlation between phenotypic and genotypic multidrug resistance should be performed.

With respect to the reviewer requirement, the present study aimed to investigate only the phenotypic resistance profile of the isolated pathogenic E. coli strains. So, the genotypic resistance investigation (the evidence of genes responsible for antimicrobial resistance) has not been performed. We highlighted this aspect as study limitation and further perspectives in this research area at the conclusion section of the revised version.

-Specific references should be added to all the used methods and techniques.

For all of the described methods and techniques in the present manuscript specific references were indicated.

-Results:

-Add this subtitle: Phenotypic characteristics of the recovered isolates

The phenotypic characteristics of the isolated strains were mentioned within the “step-by-step” presentation of the isolation and identification of the E. coli strains. However, as the reviewer suggested, the lacking of the 3.1 subheading from the original submission was corrected adding the “3.1. Prevalence of the isolated E. coli strains in camel meat samples”

  • Illustrate the phenotypic characteristics of the recovered E. coli isolates.

The following information’s were inserted in the revised version of the manuscript “Typical E. coli O157:H7 and E. coli O55:H7 isolates exhibited positive reactions for indole production and methyl red tests, as well as negative reactions for each of Voges- Proskauer and citrate utilization tests.”

-Discuss more on significant results of this study.

The significant results of the study are largely discussed in the revised version of the manuscript.

-Antimicrobial susceptibility testing:

  • Illustrate in a new table the occurrence of MDR (Multidrug resistance) among the recovered isolates as the following (illustrate the names of the antimicrobial classes and different antibiotics):

No. of strains/Type of resistance

The requested information’s regarding the multidrug resistance of the isolated strains are completely presented in the Table 3. In addition, according to the reviewer requirement, a new column was inserted in the Table 3 indicating the name of the antimicrobial classes with antimicrobial resistance.

(Antimicrobial classes and different antibiotics). The antibiotic -resistance genes

The screening of the genotypic resistance (antimicrobial resistance genes) of the isolated E. coli strains was not performed in the present study, because it was not its aim. This fact was highlighted as study limitation and further research area in the conclusion section of the revised version.

-The correlation (Correlation coefficient) between phenotypic and genotypic multidrug resistance should be performed.

The screening of the genotypic resistance (antimicrobial resistance genes) of the isolated E. coli strains was not performed in the present study, because it was not its aim. This fact was highlighted as study limitation and further research area in the conclusion section of the revised version.

-You must support your results with illustrating figures.

All of the significant results of the study, according to the authors opinion, were included in the manuscript!

-Discussion:

- Discuss more on the alarming results of this research.

All of the alarming results were deeply discussed throughout the manuscript, as highlighted by another reviewer “You make an extremely good discussion”

- The authors are advised to illustrate the real impact of their findings.

The public health impact of the study was largely approached in the revised version.

-Illustrate the different mechanisms of antimicrobial resistance in E. coli.

The study of the mechanisms of antimicrobial resistance of E. coli it was not among the aims of the present study.

Use the following valuable studies performed on pathogen isolates in the discussion section and add related references, including:

https://doi.org/10.1007/s11033-022-07215-5

In agreement with the reviewer suggestion the referred study was introduced in the reference list of the revised version of the manuscript.

-Conclusion

- Should be rephrased to be sounded. A real conclusion should focus on the question or claim you articulated in your study. Which resolution has been the main aim of your paper?

During the revision process, the authors tried to do their best to fulfill this requirement! Please see the new version of the revised conclusion section!

THANK YOU AGAIN FOR THE REVIEW!

Reviewer 3 Report

The authors aimed to investigate the prevalence of E. coli O157:H7 and E. coli O55:H7 in the marketed camel meat  distributed in camel butchers’ shops in Beheira province, Egypt, and to explore their antimicrobial resistance profile against 14 antimicrobials from different categories. Besides was investigated their prospective to cause serious infection in humans by identifying certain  virulence genes comprising eae, stx2, stx1, and hlyA genes in their genome.

The study covers some issues that have been overlooked in other similar topics. The structure of the manuscript appears adequate and well divided in the sections. Moreover, the study is easy to follow, but some issues should be improved. Some of the comments that would improve the overall quality of the study are:

I-) Authors must pay attention to the technical terms acronyms they used in the text;

II-) Please stated the limitation of the study.

Author Response

Reviewer #3:

The authors aimed to investigate the prevalence of E. coli O157:H7 and E. coli O55:H7 in the marketed camel meat distributed in camel butchers’ shops in Beheira province, Egypt, and to explore their antimicrobial resistance profile against 14 antimicrobials from different categories. Besides was investigated their prospective to cause serious infection in humans by identifying certain virulence genes comprising eae, stx2, stx1, and hlyA genes in their genome.

The study covers some issues that have been overlooked in other similar topics. The structure of the manuscript appears adequate and well divided in the sections. Moreover, the study is easy to follow, but some issues should be improved. Some of the comments that would improve the overall quality of the study are:

Dear reviewer, our sincere thanks for taking the time to review this manuscript, and your close attention to detail. We highly appreciate your overall positive feed-back regarding the quality of the manuscript! Please see below for our responses to your comments:

I-) Authors must pay attention to the technical terms acronyms they used in the text;

During the revision process, the authors tried to do their best to fulfill this requirement! In this regard, the authors hope that the possible remained non-conformities will be solved with the help of the professional MDPI team during proofreading!

II-) Please stated the limitation of the study

This reviewer request was approached within a combined action. Thus, to complete the study limitations, and to add some supplementary discussions requested by another reviewer, three solid paragraphs were inserted in the revised version of the manuscript as follows:

Discussion section

„Even if the yielded striking findings of the virulence pattern of the isolated E. coli strains in the present study can be considered suggestive, strengthening the occurrence of positive results recorded in other surveys with different detection rates, further investigations, focusing on the evidence of other virulence genes/factors (e.g. elt, est, daaD, invE, Eagg, or astA genes) enrolling a larger number of isolates are still necessary, for a comprehensive understanding of the pathogenicity of camel meat origin E. coli strains.”

and

„Altogether, the recorded overall antimicrobial resistance pattern of the isolated camel meat origin pathogenic E. coli strains can be considered worrisome in the present study, which largely agree with results of other studies carried out in other regions of the world. These results strengthen the fact that the antimicrobial resistance phenomenon became a significant threat to human and veterinary medicine. However, the recorded differences in terms of susceptibility towards the tested antimicrobials can be markedly influenced by the number of enrolled samples, study design, the used testing methodologies, and guidelines in interpreting the results, etc. Furthermore, supplementary studies, focusing on the continuous monitoring fo the presence of pathogenic E. coli strains in camel meat combined with the testing of their resistance towards a wider variety of antimicrobials are still recommended.”

Conclusion section

“However, for a better understanding of the antimicrobial resistance phenomenon in case of the camel meat origin E. coli pathogenic strains, further investigations, based on the screening of the antimicrobial resistance genes, in a larger number of samples, are still required.”

THANK YOU AGAIN FOR THE REVIEW!